# Large-scale estimation of bacterial and archaeal DNA prevalence in metagenomes reveals biome-specific patterns

Raphael Eisenhofer,[1] Antton Alberdi,[1] Ben J. Woodcroft[2]

**ABSTRACT** Metagenomes often contain many reads derived from eukaryotes, but there is usually no reliable method for estimating their prevalence. This forces many analysis techniques to make the often-faulty assumption that all reads are prokaryotic. Here, we present SingleM prokaryotic_fraction (SPF), an algorithm that scalably and robustly estimates the number of bacterial and archaeal reads in a metagenome. It also estimates the average genome size of bacteria/archaea in a sample. SPF does not use eukaryotic reference genome data and can be applied to any modern Illumina metagenome. Based on SPF, we propose the domain-adjusted mapping rate (DAMR) as an improved metric to assess prokaryotic genome recovery from metagenomes. Applying SPF to 136,284 publicly available metagenomes, we report substantial variation in prokaryotic fractions and biome-specific patterns of prokaryotic abundance, providing insights into how microorganisms and eukaryotes are distributed across Earth. Finally, we show that substantial amounts of human host DNA sequence data have been deposited in public metagenome repositories, possibly counter to ethical directives that mandate screening of these reads prior to release. As the adoption of metagenomic sequencing continues to grow, we foresee SPF being a valuable tool for the appraisal of genome recovery efforts and for investigating global patterns of microorganism distribution.

**IMPORTANCE** Metagenomics data sets capture DNA from all organisms in a sample, enabling the analysis of communities without relying on culture-based techniques. However, many samples include uncharacterized eukaryotic organisms and viral elements, meaning the proportion of bacterial and archaeal DNA is often unknown. This study presents SingleM prokaryotic_fraction (SPF), a robust and scalable method for estimating the prevalence of bacterial and archaeal DNA in metagenomes. Crucially, SPF is calculated independent of eukaryotic and viral reference genomes, which are often incomplete or unavailable. Applying SPF to over 136,000 public metagenomes uncovered substantial variability between microbial communities living in different environments. SPF also identified previously overlooked human genetic data contamination in public data sets, raising important ethical and privacy considerations. Building on SPF, we propose the domain-adjusted mapping rate (DAMR) metric, a new metric that improves genome recovery assessment by accounting for non-prokaryotic reads.

**KEYWORDS** metagenome, eukaryotic contamination, genome recovery

**Peer Reviewer** Titus Brown, University of California Davis, Davis, California, USA

Address correspondence to Raphael Eisenhofer, raph.eisenhofer@gmail.com, or Ben J. Woodcroft, b.woodcroft@qut.edu.au.

Raphael Eisenhofer and Ben J. Woodcroft contributed equally to this article.

The authors declare no conflict of interest.

See the funding table on p. 13.

Shotgun sequencing is a common technique used to study DNA extracted from microbial habitats of all kinds, both host-associated and environmental (1). In many habitats, Bacteria and Archaea (together called "microbial" or "prokaryotic" here) are numerically dominant and give rise to the majority of reads sequenced. However, for many samples, it is not clear at the outset of the experiment how much of the DNA is prokaryotic. DNA might be derived from other elements of the community such as fungi, protozoa, or viruses. In host-associated microbiomes, DNA might also be derived

from a host's genome, diet, symbionts, or parasites (2). Metagenomic analysis techniques typically assume that the fraction of reads derived from non-microbial sources is negligible. For instance, read-centric techniques ascribe functions to all reads, and tools for estimating taxonomic profiles from metagenomic data may use reference databases composed only of Bacteria and Archaea (3). This simplifying assumption is usually made as a consequence of our inability to accurately assess the quantity and identity of non-microbial reads in metagenomic data sets. How often this assumption holds is an open question.

Genome-resolved metagenomics is a strategy that is increasingly employed to study microbiomes (4, 5). In this approach, DNA sequences (typically short reads) are assembled into larger contiguous sequences (contigs), which are then sorted (binned) using nucleotide frequencies and/or read coverage information into metagenome-assembled genomes (MAGs) (6–8). This powerful approach can generate near-complete genomes from diverse sample types—providing genomic information for uncultured microbial diversity (9). Despite significant progress in laboratory and bioinformatic methodology, however, recovered genomes rarely represent the entirety of the sequenced community.

The standard metric for evaluating genome recovery efforts is calculated by mapping metagenomic reads against the recovered MAGs (10). If the vast majority of reads map, then genome recovery has been successful. In contrast, a low proportion of reads mapping can indicate that the microbial community is not well represented by the MAGs. This poor representation might be due to insufficient coverage for assembly or other challenges, such as high population heterogeneity interfering with assembly. However, the presence of non-prokaryotic reads in a metagenome from eukaryotic host or phage will also reduce read mapping rates. If non-prokaryotic reads are sufficiently numerous and not accounted for, genome recovery efforts will be falsely judged as poor by the standard metric even if they represent the entire microbial community.

When non-prokaryotic sources of DNA are known *a priori*, and high-quality reference genomes of these sources are available, filtering the corresponding reads from metagenomes is straightforward. For example, human saliva samples typically have a high proportion of host reads (>80%), which can be removed by mapping to a human reference genome (11). Indeed, removal of human reads before submission to public sequencing databases is typically considered an ethical requirement for human metagenome studies (12, 13). However, in microbiomes from other hosts and environments, removal of non-prokaryotic reads is often intractable. Even if a high-quality host genome is available, there is a lack of reference genomes for most dietary species (e.g., plants, insects, etc.) and non-microbial species to be included in a read mapping database may not be known (e.g., novel fungal diversity in soil metagenomes) (14). Therefore, to properly inform genome recovery efforts, a method to accurately estimate the proportion of prokaryotic reads in metagenomes is required. Estimation of this proportion (herein "prokaryotic fraction") would also guide researchers attempting to enrich, deplete, or maintain microbial DNA through laboratory-based cell size filtration or differential cell lysis methods (15, 16).

In this paper, we introduce SingleM prokaryotic_fraction (SPF), a tool for estimating the quantity of bacterial and archaeal DNA in metagenomic data sets and their associated genome sizes. Instead of filtering out non-prokaryotic reads, SPF bases its estimate purely on the detection of reads encoding prokaryotic single copy marker genes. An important advantage of this approach is that it does not require any knowledge of non-prokaryotic genomes such as Eukaryota or viruses, which are often missing from reference genome databases. This accurate detection of entire prokaryotic communities was recently made possible by the development of SingleM, a tool that reliably detects bacteria and archaea even when they are not represented in reference genome databases (17).

We validate SPF on simple, complex, and diverse simulated metagenomes. Two real data sets are then used to demonstrate how SPF can improve the evaluation of genome recovery efforts. Finally, we apply this fast and scalable approach to 136,284 publicly

available metagenomes to demonstrate that variation in prokaryotic fractions exists both within and between metagenomic sample types.

A side benefit of accurate prokaryotic fraction estimation from metagenomes is improved bacterial and archaeal average genome size (AGS) estimation in situations where eukaryotic DNA is present. Current tools (e.g., MicrobeCensus [18]) predict AGS by assuming that all reads are prokaryotic and ignoring the taxonomy of each read. This approach can severely overestimate AGS (154%–11,574%) when non-prokaryotic DNA is present (19). We discuss and benchmark SPF-based estimation of AGS in Notes S4 and S5.

## RESULTS AND DISCUSSION

### Benchmarking on simulated metagenomes

We first tested SPF on two simple simulated metagenomes, composed of eight bacterial and two fungal species (Zymo Mock: based on the ZymoBIOMICS Microbial Community Standard). SPF estimated the prokaryotic fractions to be 88.8% and 91.0%, very close to the 88.7% and 89.0% ground truth values (Fig. 1A). The estimated prokaryotic fractions were also close to the correct values when simulating metagenomes with varying proportions of spiked-in eukaryotic DNA (mean 3.1% ± 2.7% SD, Fig. 1A).

Next, we tested SPF on more complex simulated metagenomes: two "marine" benchmark data sets based on compositional profiles of the deep sea environment from the CAMI II challenge (20), consisting of ~400 bacterial and archaeal species alongside ~200 plasmid/viral sequences; and two "strain-madness" benchmark data sets comprising ~400 bacterial and archaeal genomes with high strain-diversity (97% of genomes have a closely related strain). SPF estimated 96.6% and 97.4% of the reads to be prokaryotic for the marine data sets (c.f. 97.7% ground truth), and 95.1% and 98.2% of the strain-madness data sets, respectively (c.f. 100% ground truth) (Fig. 1B). These results suggest that SPF is not substantially affected by high community diversity.

At the other extreme, simple communities composed mostly of novel lineages are less reliably predicted. In simple communities, SPF is highly influenced by the genome sizes of the dominant community members. When these dominant lineages are novel, their genome sizes are uncertain, which makes SPF less accurate (Note S1). Recognizing this situation, SPF emits a warning when simple communities dominated by novel lineages are encountered, a situation we found to be rare in practice (Fig. S1 and Note S1).

To test how robust SPF is when applied to complex communities where some members are novel, communities were simulated where community members were either known from GTDB R207 or new in GTDB R214 (novel relative to the R207 database used). Abundances were modeled after the CAMI 2 "marine" abundances, with 0%–100% of the community novel. SPF performed well in this benchmark, averaging 1.9% error overall from the true value (100%), and 2.8% when the community was fully comprised of novel species (Fig. 1C). Overall, *in silico* benchmarking suggests that SPF is largely accurate, apart from exceptional circumstances where simple communities contain very novel lineages.

### Benchmarking on real metagenomes

We next tested SPF on real metagenomic data to demonstrate how it can be used to evaluate genome recovery efforts.

#### *Human fecal samples*

Human fecal metagenomes usually contain ~90% microbial DNA (21) and can therefore be used as a control to benchmark SPF. After subsampling sequencing data sets to 5 Gbp per sample, genomes from 10 fecal samples (22) were recovered, yielding a total of 200 medium- or high-quality MAGs. The fraction of reads mapped against these MAGs was 46.6% ± 4.1% (Fig. 2A), much less than expected. In contrast, SPF estimates of close to 100% (96.7% ± 3.23%) agreed with our *a priori* expectations. The SPF estimates,

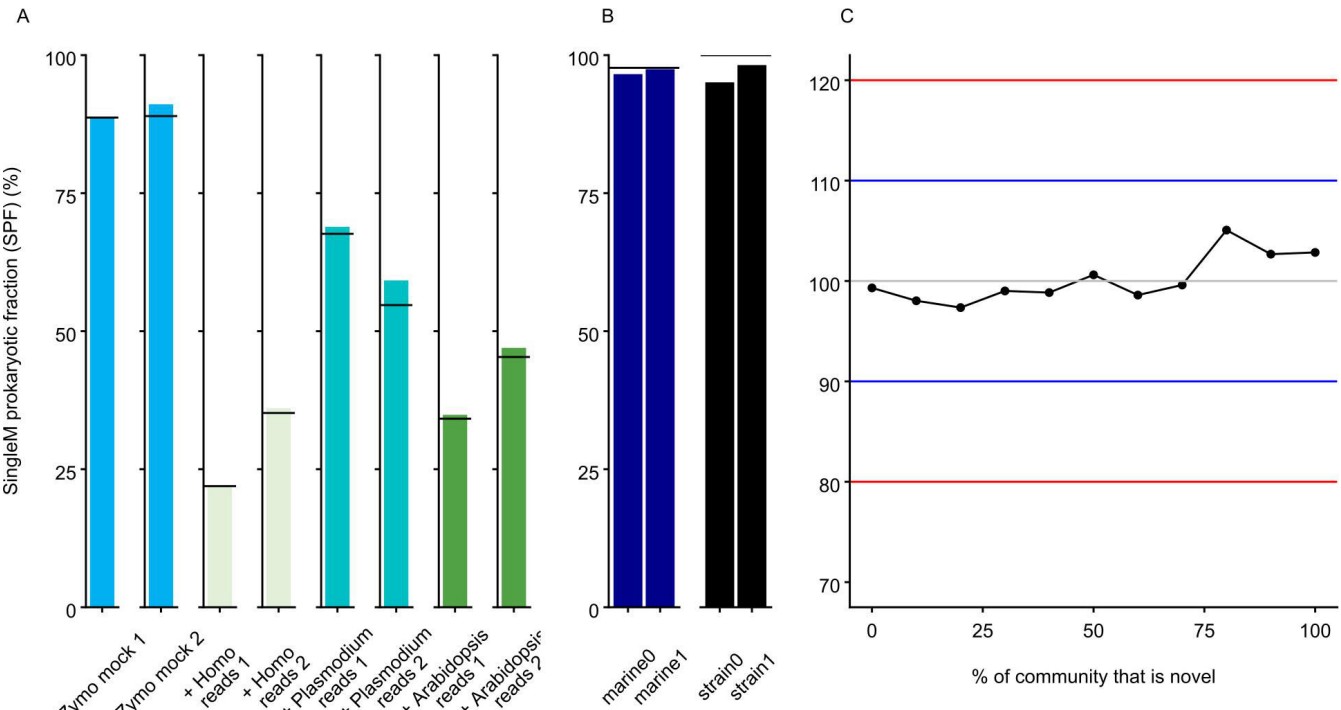

**FIG 1** Performance of SPF on simulated metagenomes. (A) Simulated eukaryotic reads from *Homo sapiens*, *Arabidopsis thaliana*, and *Plasmodium falciparum* were spiked into simulated Zymo mock community metagenomes. Horizontal bars represent the true proportion of prokaryotic DNA in the simulated metagenomes. (B) Simulated reads based on community compositions from the more complex CAMI II marine and strain-madness data sets. (C) SPF values estimated from simulated communities containing different quantities of novel lineages (lineages present in GTDB R214 but not R207, when SPF was estimated using an R207 reference database). Gray, blue, and red lines represent ground truth, +10%, and +20% SPF values, respectively. Here, the raw SPF values (dots) are shown for benchmarking clarity; when applied to real data, the tool reports a maximum of 100% SPF since larger percentages are impossible.

therefore, predicted that the low mapping rates were due to challenges in genome recovery rather than the presence of non-microbial DNA in the metagenomes. This prediction was confirmed by mapping reads to the more extensive Human Reference Gut Microbiome (HRGM) genome catalog (22), which recruited 90.4% ± 2.36% of reads. These results show that SPF can be used to attribute poor mapping rates to imperfect genome recovery when metagenomes are dominated by microbial reads.

### Hyena fecal samples

Unlike humans, non-model species often lack both host reference genome sequences and appropriate microbiome genome catalogs. They may contain varying quantities of DNA derived from dietary and/or protozoan sources (23–25). Hyenas are carnivores with diets that can fluctuate throughout the year; hence, we expected that fecal samples could contain varying proportions of prey DNA from their diets. This variability can make it difficult to determine whether the microbial diversity in a sample is adequately captured using standard assessments. We applied SPF to a recent study on wild spotted hyenas (*Crocuta crocuta*)—the first metagenomic study for this species (26). Each hyena was sampled multiple times; hence, we coassembled metagenomic reads from hyena fecal samples by individual and binned contigs into *de novo* MAGs. Our reprocessing improved the mappability of each sample's reads to the *de novo* MAGs substantially compared to the original study (original study: 13.2% ± 7.7%, our study: 52.8% ± 15.5% SD).

SPF estimated substantial variation in the proportions of microbial DNA in the hyena samples (20.2%—100%), unlike in the human fecal samples (Fig. 2B). The proportion of reads recruited to MAGs was roughly proportional to SPF, providing a level of validation for the SPF values. In all cases, the SPF was higher than the percentage of reads mapped

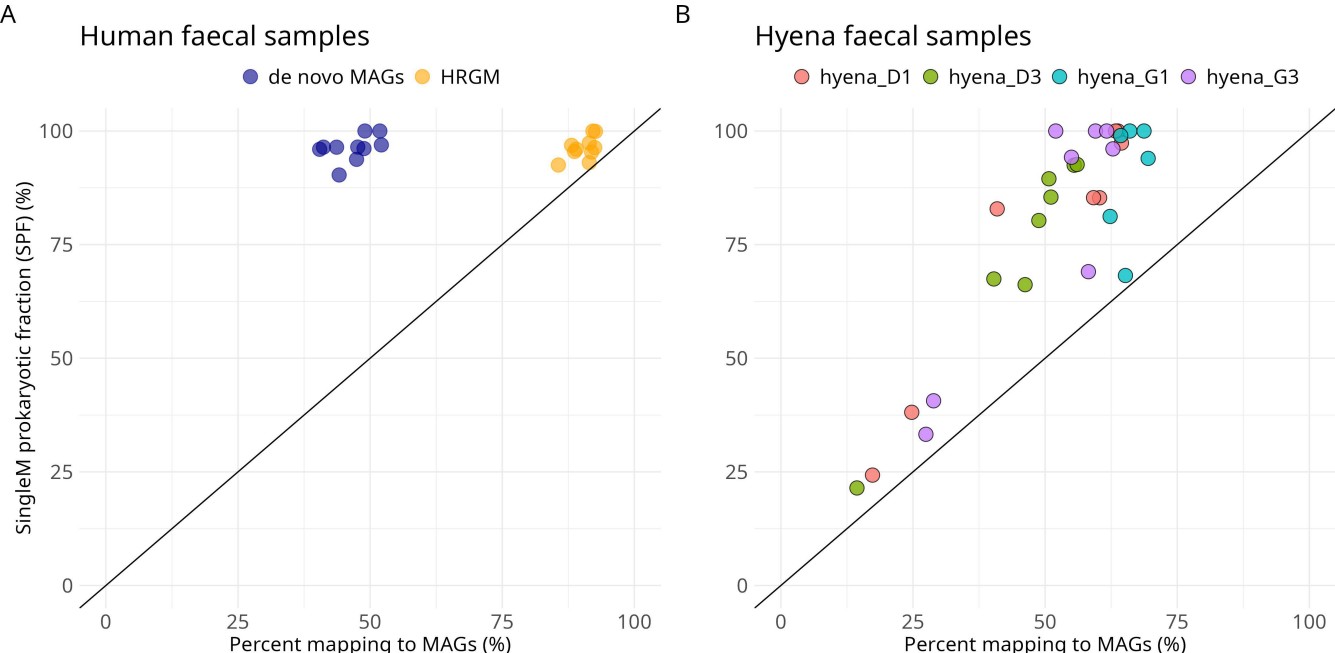

**FIG 2** SPF can help evaluate genome recovery efforts. (A) Human fecal data set. Percentage of reads mapping to the *de novo* MAGs (blue) versus the percentage of the metagenomes estimated to be microbial by SPF. Here, the *de novo* dereplicated MAGs underrepresent the true diversity in the samples, whereas most reads are recruited to the HRGM (yellow). (B) Hyena fecal data set. Percentage of reads mapping to the MAG catalog versus SPF estimates. Points on or close to the straight line indicate that the *de novo* MAGs were sufficient to capture most of the diversity in the sample.

to the MAGs as might be expected since MAG recovery is unlikely to be perfect. However, some raw, uncapped SPF values were >100% due to the presence of a novel Bacteroides species with an unusually small genome size (Note S2).

## Domain-adjusted mapping rate

Application of SPF to these human and hyena fecal samples demonstrates that mapping rates alone can be a very poor estimation of MAG recovery success when the number of microbial reads is unknown. To overcome this limitation, here, we introduce a new metric named domain-adjusted mapping rate (DAMR). DAMR is calculated as the rate of read mapping to sets of microbial genomes (MAGs or other references) divided by the fraction of the community predicted to be bacterial or archaeal (here with SPF). For example, if a metagenome is predicted to have 50% microbial DNA, and 45% of reads map to the MAG catalog/references, the DAMR is 90% (45%/50%). In rare circumstances where the SPF estimate is lower than the fraction of mapped reads, indicating artifacts in mapping and/or SPF, then the DAMR is rounded down to 100%. In the human analysis outlined above, the DAMR of the Human Reference Gut Microbiome (HRGM) genome catalog was calculated to be 93.5%. In the hyena example, combining SPF estimates with the proportion of reads mapping to the *de novo* MAGs yielded DAMR values ranging from 49.3% to 95.6%, which allowed us to identify samples that would require additional sequencing to improve genome recovery efforts. In summary, combining SPF with the DAMR metric can be a valuable tool for interpreting the performance of prokaryotic genome recovery from understudied host species and complex metagenomic mixtures.

## Microbial read fractions of Earth's metagenomes

To explore the global trends in prokaryotic fractions among samples sourced from a broad range of environments, we applied SPF to publicly available metagenomic data sets. In total, SPF was calculated from 136,284 community profiles available on the Sandpiper (17) website. These estimates were subsequently incorporated back into Sandpiper, where SPF values are now shown for each metagenome.

SPF estimates were compared to those produced by STAT (27), a kmer-based tool for taxonomic assignment of reads. The tool is trained on RefSeq genomes and is widely available for metagenomes at the NCBI SRA. We hypothesized that because STAT is reliant on matching kmers to reference genome data, SPF would provide better estimates for understudied sample types or host species. SPF and STAT estimates were within ±5% of each other for 18.2% of samples. SPF yielded higher prokaryotic fraction estimates (at least >=5%) than STAT for 58.6% of metagenomes (Fig. 3A). In 33.2% of the samples, the SPF estimation was >2-fold higher than STAT, consistent with our hypothesis that STAT underestimates metagenomes containing novel lineages. Conversely, in only 7.1% of metagenomes was STAT >2-fold higher than SPF (28.5% of these were "viral metagenomes").

We next investigated samples from different environments. For human fecal metagenomes, we restricted the analysis to metagenomes with location information curated by Martiny et al. (28) ($n$ = 18,571, Fig. 3B and 4). As expected, human gut metagenomes typically showed a high prokaryotic fraction (mode ~90%) (Fig. 4). SPF estimates were somewhat higher than STAT (SPF/STAT: median 77.2%/69.7%, SD

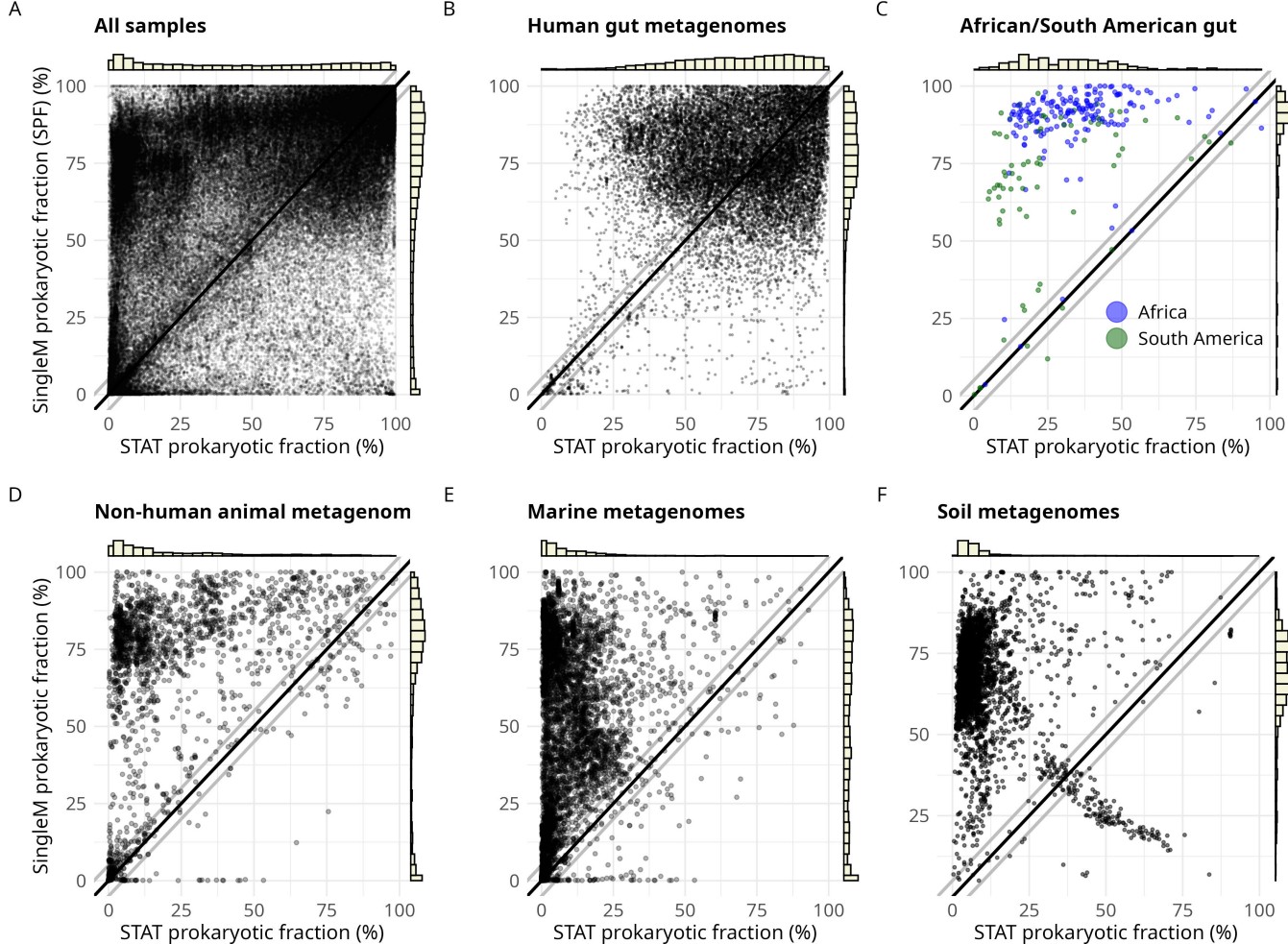

**FIG 3** Comparison of prokaryotic fractions derived from STAT and SPF. SPF (Y-axes) was run on 136,284 metagenomes from the SRA and compared to STAT prokaryotic fraction estimates (X-axes). (A) All samples. (B) 18,571 metadata-curated human gut metagenomes (warning for 0.19% of samples). (C) African and South American human fecal samples (warning for no samples). (D) In total, 2,073 curated, non-human samples (warning for 0.10% of samples). (E) In total, 5,490 curated marine metagenome samples (warning for 0.79% of samples). Note that most marine samples have been subjected to size filtration, such as some selecting prokaryotic or eukaryotic cells preferentially (see Fig. S3 for more information). (F) In total, 4,095 soil metagenome samples (warning for 0.17% of samples). Black lines represent equal SPF and STAT values, and gray lines represent ± 5%. The bar charts represent the distributions of SPF and STAT values. The small cluster of samples with a higher STAT estimate than SPF below the diagonal is discussed in Note S3.

18.5%/21.3%), suggesting that most human fecal samples are well represented by reference databases. However, SPF estimates were substantially higher than STAT in samples derived from understudied human populations: African (median 91.8% vs 34.0%, Mann-Whitney *P*-value 3.68e-51) and South American (median 74.9% vs 21.4%, Mann-Whitney *P*-value 2.21e-15) populations (Fig. 3C). One plausible explanation is the underrepresentation of African and South American gut microbial genomes in the public databases. Most work on the human gut microbiome, to date, has been from individuals from the Global North (~71% of human fecal samples are from Europe, U.S.A., or Canada [29]).

Genomes derived from non-human animal microbiomes are also considerably underrepresented in public databases. To explore this, we used the set of SRA samples curated from the AnimalAssociatedMetagenomeDB (30) (*n* = 2,073 samples). SPF estimates were again higher for these samples (median = 76.5 SD = 31%) compared to STAT (median = 12.4 SD = 23.9%, Fig. 3D), consistent with the hypothesis that SPF is more capable of estimating prokaryotic fractions from metagenomic samples of understudied hosts. This was also the case for the subset of better-studied domestic animals (chicken *n* = 352: SPF median 89.5%, SD 12.7%; STAT median 42.9%, SD 19.6%; and Bovine *n* = 202: SPF median 78.4%, SD 22.1%; STAT median 7.4%, SD 5.6%).

Marine metagenomes are derived from the largest biome on Earth, with samples sourced from varying geographic locations, depths, and sample types (e.g., water and sediment). SPF yielded much higher prokaryotic fraction estimates than STAT (Fig. 3E) as expected, given the paucity of RefSeq quality reference genomes from marine environments (31). Marine water metagenomes are typically generated by filtering seawater through pores of a defined size (16). The size of these pores varies between studies, where ultrasmall (<0.22 µm) size selection targets viruses, intermediate (0.22–3 µm) sizes target Archaea and Bacteria, and larger sizes (3 µm+ or 20 µm+) enrich for eukaryotic plankton (16). Marine metagenome SPFs were strongly influenced by filtration size, with ultrasmall (17.7% ± 15.0%) and larger sizes (3.7% ± 6.1%) exhibiting a much smaller SPF than intermediate sizes (70.6% ± 13%, Fig. S3). We also used SPF values to assess the performance of recent efforts to create a unified marine genome reference catalog for marine metagenomes (31). The estimated DAMR for the Unified Genome Catalog of Marine Prokaryotes (UGCMP) was 63.4% (SD 13.6%) (Fig. S4), suggesting that many marine microbial genomes are yet to be recovered.

Soil metagenomes were found to contain mostly microbial DNA, with 88% having SPF >50% (median 69.0%, SD 15.1%, Fig. 4). Soil microbial communities are poorly

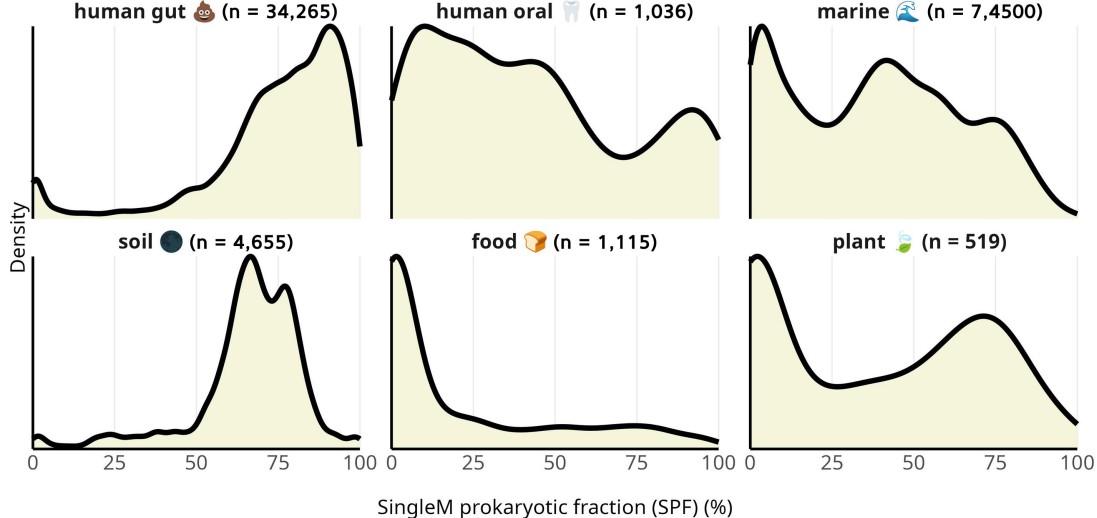

**FIG 4** SPF of metagenome sample types sourced from the SRA. Note that the marine samples include those derived from filtration at various sizes (e.g., some selecting prokaryotic or eukaryotic fractions preferentially), a phenomenon investigated separately (Fig. S3).

represented in genome databases (17, 32); hence, STAT estimates were almost always much lower (14% were >50%, and 68% were <10%, Fig. 3F; Note S3). These analyses confirm SPF's ability to estimate prokaryotic fraction in metagenomes containing many novel species. They also show that the well-known challenges for recovering microbial genomes from soil (32–34) are not usually a result of eukaryotic DNA dominating the sample.

Soils harbor many different kinds of organisms, with bacteria and fungi thought to be the most dominant members overall. The ratio of fungi to bacteria, however, can be variable. Using quantitative amplicon-based sequencing, Fierer et al. observed ratios of 0.007 to 0.34 among a range of unmanaged soil environments (35). Under some simplifying assumptions (see methods), the median SPF value (69.0%) corresponds to a fungal to microbial cell ratio of 0.053. The 90th and 10th SPF percentiles for soil (82% and 48%) correspond to fungi:bacteria cellular ratios of 0.03 and 0.13, in rough agreement with Fierer et al. (35).

Overall, metagenomes derived from different environments had substantially different prokaryotic fractions, and individual environment types were often heterogeneous (Fig. 4). For instance, samples labeled "human oral metagenome" had a wider distribution of prokaryotic fractions, which likely relates to the diversity of distinct oral niches (Fig. 4) (36). Saliva samples typically have high proportions of host DNA (>80%), whereas tongue swab samples have lower proportions (~10%) (21). Most of the "food metagenome" samples had low microbial read fractions (median 9.2%), highlighting potential issues inherent with these sample types, such as high quantities of raw ingredients. For instance, we found cow DNA in milk SRA (run ERR3143475 [37] had SPF 1% and 76% of reads assigned to the family Bovinae according to STAT) and likely fungal DNA in ferments (SRR648391, a metagenome derived from soy sauce, had SPF 24%). Plant metagenomes exhibited a distribution of different prokaryotic fractions, likely related to variation in the types of samples in these categories (e.g., phyllosphere and rhizosphere) (31, 38). These results highlight that prokaryotic fractions can differ both within and between biomes.

Raw metagenomes derived from human microbiomes contain substantial quantities of human DNA, particularly from non-fecal samples. However, since making human genetic data public might result in invasions of privacy, it is common practice to screen out human host DNA before submitting the data to publicly available repositories (39). We were therefore surprised to find many human microbiome samples with low SPF values (e.g., in human oral metagenomes; Fig. 4). After confirming many of these samples did indeed contain human DNA using STAT, we were forced to conclude that public metagenome databases contain large amounts of human genetic data. While metagenomic samples are usually deidentified, the availability of this data could have ethical and practical implications in some cases (13, 40).

## Conclusion

SPF can be utilized as a scalable and assembly-free means to guide researchers' decision-making by prioritizing samples prior to deeper sequencing, assessing microbial community representativeness, as well as benchmarking bioinformatic genome-recovery pipelines and laboratory method development. It will be especially useful as the field continues to explore novel sample types and hosts with unknown fractions of microbial DNA. Finally, the microbial read fractions generated for a large number of public metagenomes provide a new fundamental variable that can be used to explore how experimental or environmental variables shape microbial communities.

## MATERIALS AND METHODS

### Implementation of SingleM prokaryotic_fraction (SPF)

The SingleM prokaryotic_fraction (SPF) workflow is implemented as a post-processing step (implemented in the singlem prokaryotic_fraction mode) of community profiles generated by the main workflow of SingleM (singlem pipe) (17), which estimates community profiles from unassembled metagenomic reads. The SPF approach is presented here as a post-processing of SingleM-generated community profiles, but SPF can also be applied to community profiles generated by other metagenome-based profiling tools, so long as they (i) estimate lineage-wise read coverage rather than lineage-wise read percentage and (ii) incorporate novel species into their profiles. The input to SPF is a "SingleM condensed" format file to specify the community profile (a simple sparse format readily generated from other community profile formats) and a count of total bases in the metagenome (alternately, if the metagenome itself is supplied, SPF will count the bases).

The main workflow of SingleM takes as input raw metagenomic reads and uses 59 hidden Markov models to search for reads matching conserved sections of single-copy core genes. Taxonomy and count of recruited reads are then harmonized between genes, resulting in a predicted community composition based on Genome Taxonomy Database (GTDB) (41) taxons. The per-base read coverage of each taxon is also estimated. For instance, if a culture of a species that has a 1,000,000 bp genome is sequenced yielding 1,000,000 reads that are each 150 bp long, then the coverage of that species is 150, since on average each position on the genome will be covered by 150 reads on average.

The community profile output by SingleM provides the information required to estimate the relative abundance of each taxon in a community. However, the per-base read coverage can also be used to estimate the total number of reads in the metagenome belonging to that taxon. This is what SPF does.

At its core, SPF operates by applying a simple equation for each species similar to the Lander/Waterman equation (42):

$$\text{bases\_covered} = \text{coverage} \times \text{genome\_size}$$

where coverage is the average number of reads covering each base in the species' genome. Here, we use SingleM's estimate of the coverage and assume a single genome_size for each species. We then estimate the total number of bases in the metagenome that derive from all microbial genomes by summing the estimates for each species:

$$\text{microbial\_bases\_in\_metagenome} = \sum_{\text{species\_i}}^{\text{microbial\_species}} \text{bases\_covered}_{\text{species\_i}}$$

We finally estimate the fraction of the metagenome's reads that are prokaryotic, with a maximum of 100%:

$$\text{SPF} = \min\left\{\frac{\text{microbial\_bases\_in\_metagenome}}{\text{bases\_in\_metagenome}} \times 100\%, \ 100\%\right\}$$

### Estimation of genome size for SPF calculation

To estimate the true genome size of each species, accounting for imperfections that arise from genome recovery efforts, we take the genome size of the GTDB species representative and adjust it based on its completeness and contamination as predicted by CheckM2 (43), where completeness and contamination are expressed as decimal numbers between 0 and 1:

$$genome\_size\_adjusted \; = \; \frac{genome\_size}{completeness \; \times \; (1 \; + \; contamination)}$$

Community profiles generated by SingleM also contain entries from lineages not resolved to the species level, for example, coverage assigned to the genus or family level. For these cases, we use genome sizes estimated from lower taxonomic ranks. For genera:

$$genus\_genome\_size \; = \; mean(species\_genome\_sizes\_adjusted)$$

where species_genome_sizes_adjusted is the set of adjusted genome sizes of species within the genus. For family, order, class, phylum, and domain levels, we average the average genome sizes of the taxons immediately below them. For families:

$$family\_genome\_size \; = \; mean(genus\_genome\_sizes)$$

where genus_genome_sizes is the set of genome sizes of genera belonging to that family. Orders, classes, phyla, and domains are calculated similarly based on the mean genome sizes of their families, orders, classes, and phyla, respectively.

## Warnings about unreliability of estimates for some communities

Some communities are dominated by individual lineages that are not currently represented at the species level in reference genome databases. These communities are challenging for the approach used here because incorrect genome size estimation(s) can have a large impact on SPF. In simple communities composed of a few genomes, the SPF value can vary markedly with the (uncertain) estimate of only one or two lineages' genome sizes (Note S1).

In more complex communities such as those found in soils and many animal guts, the genome sizes of individual lineages may also be uncertain. However, in these cases, SPF is derived from many independent estimates of genome size (one for each lineage), even if the individual estimates are uncertain. For SPF to be inaccurate, many genome size estimates have to be incorrect and either systematically overestimated or systematically underestimated. Therefore, complex communities may be less affected by uncertain genome size estimates.

Following this observation, singlem prokaryotic_fraction emits a warning when simple communities with novel species are encountered. Specifically, we consider the three most abundant lineages not classified to the species level in the profile. If doubling or halving their estimated genome size changes the SPF by more than 2%, then a warning is emitted.

Users can improve the accuracy of SPF by enriching the SingleM reference database with user-generated MAGs, that is, producing an updated SingleM "metapackage." This can be achieved by using singlem supplement to add the new MAGs, calculate community profiles using this metapackage with singlem pipe, and then use singlem prokaryotic_fraction to generate updated SPF values. An example of this workflow is provided in Note S2.

## Estimation of average microbial genome size in a metagenome

Average microbial genome size in a metagenome is calculated by singlem prokaryotic_fraction using the per-taxon genome sizes estimated according to the procedure above. The average genome size is the average of these genome sizes after weighting them by their coverage in the SingleM community profile. That is, it is calculated according to:

$$average\_genome\_size \; = \; \frac{\sum_i^{taxons} genome\_size_i \times coverage_i}{\sum_i^{taxons} coverage_i}$$

The approach used here differs from previous methods at estimating average genome size, which assume that the entire metagenome is derived from microbial genomes (44, 45).

## Benchmarking using simulated data

We benchmarked the performance of SPF on four simulated data sets. Gen-paired-end reads (https://github.com/EisenRa/reads-for-assembly) (forked from https://github.com/merenlab/reads-for-assembly) were used to generate simulated paired-end reads from the Zymo Mock Community Standard reference genome files (ZymoBIOMICS.STD.refseq.v2; also available on our GitHub repository), host reference genomes (*Homo sapiens*: GCF_000001405.39_GRCh38.p13; *Arabidopsis thaliana*: GCF_000001735.4_TAIR10.1; *Plasmodium falciparum*: GCF_000002765.5_GCA_000002765), and CAMI II marine and strain-madness reference genomes (profile #0 for each CAMI II data set) (20). For more information and reproducible code, see https://github.com/EisenRa/SingleM_microbial_fraction_paper.

In a separate simulation to benchmark the performance of SPF on communities including divergent species (Fig. S1), we used a Snakemake (46) workflow available at https://github.com/wwood/singlem-read-fraction-benchmarking. Reads were simulated for 120 genomes from species that were added to the GTDB database in R214. Reads were simulated using ART (47) at 10× coverage of two microbial genomes: one new in R214 and the other present in R207. The pair of genomes in each mock community was chosen to have similar genome sizes and such that one was bacterial and the other archaeal. SingleM v0.16.0 was run first in "pipe" mode using the S3.1.0 metapackage (48) to generate a GTDB R207 community profile from the simulated reads. The microbial_fraction mode was used to apply SPF using the same metapackage but an updated taxon-genome-lengths file available in the benchmarking "singlem-read-fraction-benchmarking" GitHub repository linked above.

MicrobeCensus v1.1.1 (18) average genome size estimates were generated using default parameters on the simulated Zymo mock communities.

## Analysis of previously published data

Sequencing reads from human (22) and spotted hyena (26) fecal samples were downloaded using Kingfisher (49). Ten random human fecal samples were selected and randomly subsampled to 5 Gbp of data using seqtk "sample" (https://github.com/lh3/seqtk). Hyena samples from four individuals (eight samples per individual) were used, and host reads were removed by mapping to the spotted hyena reference genome (GCA_008692635.1) using BowTie2 (50). SingleM version 0.16.0 was used to estimate prokaryotic fractions with metapackage 3.2.1 (https://zenodo.org/records/8419620), which was based on GTDB (41) R214. For both data sets, the Earth Hologenome Initiative (EHI) genome-resolved metagenomic pipeline was used (https://github.com/earthhologenome/EHI_bioinformatics) to recover MAGs. Briefly, the reads were quality-trimmed using fastp (51) and mapped to their respective host genomes using Bowtie2 (50), before removing mapped reads with samtools (52). Unmapped reads were then co-assembled using megahit (6), before being binned and refined using MetaWRAP's binning (MetaBAT2 [53], MaxBin2 [8], and CONCOCT [7]) and refinement modules (54, 55). The resulting bins were dereplicated at 98% ANI using dRep (56), and the processed reads were then mapped to their respective MAG catalogs using Bowtie2. The final count tables were created using CoverM (v0.6.1) (57) "genome" using --min-covered-fraction 0.

## SPF estimates across public metagenome data sets

To calculate the SPF across public metagenomes, taxonomic profiles from Sandpiper 0.2.0 were taken as input to "singlem microbial_fraction" v0.16.0 (the mode has since been renamed to "singlem prokaryotic_fraction") using the --input-metagenome-sizes option, supplying metagenome sizes derived from the NCBI SRA BigQuery SQL database

(column "mbases"). Samples were filtered by SRA metadata: library_strategy == "WGS" and library_selection == "RANDOM." Only samples with > 0.5 Gbp were analyzed. As the SRA can contain samples with incorrect metadata entries, for some analyses, we opted for the usage of samples that had been manually curated. We used samples with curated metadata from the AnimalAssociatedMetagenomeDB (30) and MarineMetagenomeDB (58) for Fig. 3D and E, respectively. Curated location metadata from the study by Martiny et al. (28) were used for human fecal samples in Fig. 3B and C.

For soil metagenomes, metagenomes with very low SPF values (<5%) were excluded from statistical analysis. Manual inspection of a random selection of these data sets showed that they were not standard metagenomes, that is, they were single-cell amplifications or targeted at mitochondria. Metagenomes that had no annotated latitude or longitude were also excluded.

STAT-based estimates (27) of bacterial and archaeal read counts for each metagenome were downloaded on Dec 15, 2022 from the nih-sra-datastore.sra_tax_analysis_tool.tax_analysis table available in Google BigQuery, collecting the "total_count" column where the tax_id was 2 (Bacteria) or 2157 (Archaea), and the sample was estimated to be a metagenome using the same SQL parameters as SingleM. The prokaryotic fraction estimated by STAT was calculated as the sum of the bacterial and archaeal counts divided by the total number of reads in the library.

## Estimating DAMR for marine metagenomes

Mapping rates to different marine genome catalogs were obtained from Nishimura et al. (File S1) (31).

## Estimation of fungi:microbial cell ratios in soil metagenomes

To convert an SPF value—defined as a percentage of DNA reads that are microbial—into an estimate of the relative abundance of fungal and microbial cells, the following simplifying assumptions were made.

1. The community is largely composed of fungi and bacteria only, with the remainder contributing very few reads.
2. The metagenomic data are derived from DNA extraction methods that lysed fungal and bacterial cells with equal efficiency.
3. The abundance-weighted average size of fungal genomes in each sample is 37.5 Mbp. This average was derived by taking the median genome size of each *Ascomycota* (59) genus and then taking the mean of those median values. The Ascomycota phylum was chosen since it contains the most species, a large number of which have available genome sequences.
4. The average number of genome copies per cell (including in multinucleated fungal cells) is the same between fungi and bacteria.

The average genome size of microbial community members was estimated for each metagenome with the method described above. To calculate cellular ratio from the SPF, we used the following formula, where $c_m$ is the number of microbial cells, $c_f$ is the number of fungal cells, $g_m$ is the average microbial genome size, and $g_f$ is the average fungal genome size.

$$\frac{c_m}{c_f} = \frac{\text{SPF}}{\frac{g_m}{g_f}(1 - \text{SPF})}$$

This equation was derived starting from the following, where $r_m$ is the number of reads derived from microbial cells, $r_f$ is the number of reads derived from fungal cells, and $s$ is the average number of reads generated that start at each genomic position:

$$\text{SPF} = \frac{r_m}{r_m + r_f}$$

$$r_m = c_m g_m s$$

$$r_f = c_f g_f s$$

Taking the ratio of the last pair of equations, multiplying it by $r_f$ and substituting into the first equation leads to the formula above for $\frac{c_m}{c_f}$.

## ACKNOWLEDGMENTS

We thank Simon Roux and Matthew Sullivan for helpful discussion on marine data sets, as well as Joshua Mitchell for helpful suggestions for interpreting marine filter size metadata.

B.J.W. was supported by Australian Research Council Future Fellow (#FT210100521) and Discovery Project (#DP230101171) grants. A.A. acknowledges the Danish National Research Foundation award DNRF143 "A Center for Evolutionary Hologenomics" and the Carlsberg Foundation grant CF20-0460.

## AUTHOR AFFILIATIONS

[1]Center for Evolutionary Hologenomics, Globe Institute, University of Copenhagen, Copenhagen, Denmark
[2]Centre for Microbiome Research, School of Biomedical Sciences, Queensland University of Technology (QUT), Translational Research Institute, Woolloongabba, Queensland, Australia

## AUTHOR ORCIDs

Raphael Eisenhofer http://orcid.org/0000-0002-3843-0749
Antton Alberdi http://orcid.org/0000-0002-2875-6446
Ben J. Woodcroft http://orcid.org/0000-0003-0670-7480

## FUNDING

| Funder | Grant(s) | Author(s) |
| --- | --- | --- |
| Department of Education and Training | FT210100521 | Ben J. Woodcroft |
| Department of Education and Training | DP230101171 | Ben J. Woodcroft |
| Danmarks Grundforskningsfond | DNRF143 | Antton Alberdi |
| Carlsbergfondet | CF20-0460 | Antton Alberdi |

## AUTHOR CONTRIBUTIONS

Raphael Eisenhofer, Conceptualization, Data curation, Investigation, Methodology, Project administration, Resources, Software, Validation, Visualization, Writing – original draft, Writing – review and editing | Antton Alberdi, Conceptualization, Funding acquisition, Investigation, Project administration, Validation, Writing – review and editing | Ben J. Woodcroft, Conceptualization, Data curation, Funding acquisition, Investigation, Methodology, Project administration, Resources, Software, Supervision, Validation, Visualization, Writing – original draft, Writing – review and editing

## DATA AVAILABILITY

SingleM is available at https://github.com/wwood/singlem. Code and data for reproducing the analyses and figures of this paper are available at https://github.com/EisenRa/SingleM_microbial_fraction_paper and https://github.com/wwood/singlem-read-fraction-benchmarking.

## ADDITIONAL FILES

The following material is available online.

### Supplemental Material

**Supplemental Information (mSystems01062-25-s0001.docx).** Notes S1-S5, Tables S1 and S2, and Figures S1-S6.

### Open Peer Review

**PEER REVIEW HISTORY (review-history.pdf).** An accounting of the reviewer comments and feedback.

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
