## [Reviewer comments · mSystems]

Large-scale estimation of bacterial and archaeal DNA prevalence in metagenomes reveals biome-specific patterns

Raphael Eisenhofer, Antton Alberd, and Ben Woodcroft

Corresponding Author(s): Ben Woodcroft, Queensland University of Technology

Review Timeline:

Submission Date:	July 15, 2025
Editorial Decision:	September 1, 2025
Revision Received:	November 12, 2025
Accepted:	December 21, 2025

Editor: Sean Gibbons

Reviewer(s): Disclosure of reviewer identity is with reference to reviewer comments included in decision letter(s). The following individuals involved in review of your submission have agreed to reveal their identity: Titus Brown (Reviewer #1)

Transaction Report:

DOI: <https://doi.org/10.1128/msystems.01062-25>

Re: mSystems01062-25 (Large-scale estimation of bacterial and archaeal DNA prevalence in metagenomes reveals biome-specific patterns)

Dear Dr. Ben Woodcroft:

Revision Guidelines

Sincerely,
Sean Gibbons
Editor
mSystems

Reviewer #1 (Comments for the Author):

This well written and presented paper uses a new metric, "SingleM Microbial Fraction", to estimate the proportion of a metagenome that belongs to bacterial and archaeal species. This works by using the SingleM to estimate the taxonomic composition of a metagenome, and then uses the estimated coverage and genome size of each species representative to estimate the fraction of the metagenome that belongs

to bacteria and archaea. This is an intuitively pleasing and straightforward to calculate metric.

My biggest challenge with this paper is the authors' use of the term "microbial" to mean "Bacterial and Archaeal." This is really non-standard and it renders the paper difficult to read, because of course there are many microbial eukaryotes and so the language always needs to be mentally adjusted. I understand why the authors did it but I think it needs to be removed. I don't have a strong recommendation for what to replace it with.

(You can clearly see the confusion caused by this in the opening sentences of the abstract, which first states that "Metagenomes often contains many reads derived from eukaryotes" and then points out that there is no way to estimate the prevalence of non-microbial reads, which already assumes that "non-microbial" means "eukaryotic.")

Other than this, I really like the paper over all. The math presented is straightforward, and the discussions of the drawbacks and challenges of the metric is well presented. I do think the authors should mention viruses (although we can assume that non-organismal/viral reads would generally be relatively minimal, given their small genomes).

(The overall comparisons and observations seem good and interesting to me, but I am mostly focused on the SMF metric.)

Minor concerns -

line 123, "taxonomy of each reads" - each read? Or is something else meant here?

line 179, do you mean microbiomes from non-model species often lack microbial reference genomes?

Reviewer #2 (Comments for the Author):

This manuscript introduces SMF, an algorithm that aims at estimating bacterial and archaeal reads from metagenome samples without the use of eukaryotic and viral reference genomes. Useful applications of this tool will likely include underrepresented microbiomes, including those from human hosts from the Global South as well as from non-human hosts and environments. The tool should give the user an idea of the fraction of microbial reads in a given metagenome sample which can be extended to calculate the fraction of microbial reads mapped to MAGs to help understand MAG recovery success (DAMR). While SMF provides a solution to a particular problem in the microbiome field, its specific nature may make it less accessible to the average microbiome researcher. However, it is an important step in filling research gaps pertaining to underrepresented microbiomes in particular.

The paper is well-written, concise, and contains benchmarking steps that would be expected with this type of research. Benchmarking is done on a wide range of datasets, from simple mock communities, to non-human animal microbiomes and environmental samples. However, given the inherent challenges of testing metagenomic samples in the absence of a clear ground truth, the manuscript could benefit from further expanding the benchmarking in simulated metagenomes. For instance, the rationale behind the specific percentage of contaminant reads spiked in for Figure 2A is not clearly motivated, and it would strengthen the analysis to test multiple percentages. In addition, running multiple, independent simulation replicates for each data point and setting (in both Figures 2A and 2C) would improve the statistical robustness of the results and provide greater confidence in the reported performance.

The authors highlight instances where their tool is likely to give false estimates (e.g., microbiomes containing highly-abundant novel lineages or when genome size is vastly different than expected). In these cases, they have assured that the tool will warn the user of its potential pitfalls. The authors also made it clear that their tool will not necessarily provide better results than existing tools (e.g., STAT) for human fecal samples from well-studied populations. For this comparison, a short statement that the authors' results are not necessarily surprising is warranted since this approach is quite different from mapping to reference genomes.

For future users, it would be beneficial for the authors to clearly state early on in the manuscript what the input formats should be for their tool. Does SMF only work in integration with SingleM? Highlighting how dependent this tool is on pre-existing tools from the authors would be beneficial so that users can understand how easily it can be integrated into their existing pipelines.

Minor comments:

The authors can consider adding some numbers to support their statement in lines 318-319.

The authors can consider merging the paragraph in lines 262-266 with that of lines 267-278.

Lines 292 and 293 both contain "see methods". Consider keeping only one such instance.

Figure 1 is not the strongest figure and its message can easily be conveyed in the text. To remove or strengthen it is left to the authors.

We thank both reviewers for their positive and constructive feedback on the manuscript. Below, a point by point discussion of each point is included.

Reviewer #1 (Comments for the Author):

R1.1) This well written and presented paper uses a new metric, "SingleM Microbial Fraction", to estimate the proportion of a metagenome that belongs to bacterial and archaeal species. This works by using the SingleM to estimate the taxonomic composition of a metagenome, and then uses the estimated coverage and genome size of each species representative to estimate the fraction of the metagenome that belongs to bacteria and archaea. This is an intuitively pleasing and straightforward to calculate metric.

My biggest challenge with this paper is the authors' use of the term "microbial" to mean "Bacterial and Archaeal." This is really non-standard and it renders the paper difficult to read, because of course there are many microbial eukaryotes and so the language always needs to be mentally adjusted. I understand why the authors did it but I think it needs to be removed. I don't have a strong recommendation for what to replace it with.

(You can clearly see the confusion caused by this in the opening sentences of the abstract, which first states that "Metagenomes often contains many reads derived from eukaryotes" and then points out that there is no way to estimate the prevalence of non-microbial reads, which already assumes that "non-microbial" means "eukaryotic.")

The name microbial_fraction has been replaced by prokaryotic_fraction, and the SMF acronym is now SPF. These changes reduce the confusion since "prokaryotic" is a more precise term. However, we continue to use the word "microbial" in some text of the manuscript, given the broad usage of this term in the literature.

R1.2) Other than this, I really like the paper over all. The math presented is straightforward, and the discussions of the drawbacks and challenges of the metric is well presented. I do think the authors should mention viruses (although we can assume that non-organismal/viral reads would generally be relatively minimal, given their small genomes).

(The overall comparisons and observations seem good and interesting to me, but I am mostly focused on the SMF metric.)

We thank the reviewer for their positive feedback.

Minor concerns -

R1.3) line 123, "taxonomy of each reads" - each read? Or is something else meant here?

This has been corrected.

R1.3) line 179, do you mean microbiomes from non-model species often lack microbial reference genomes?

We meant both. The sentence has been reworded to make this clearer.

Reviewer #2 (Comments for the Author):

R2.1) This manuscript introduces SMF, an algorithm that aims at estimating bacterial and archaeal reads from metagenome samples without the use of eukaryotic and viral reference genomes. Useful applications of this tool will likely include underrepresented microbiomes, including those from human hosts from the Global South as well as from non-human hosts and environments. The tool should give the user an idea of the fraction of microbial reads in a given metagenome sample which can be extended to calculate the fraction of microbial reads mapped to MAGs to help understand MAG recovery success (DAMR). While SMF provides a solution to a particular problem in the microbiome field, its specific nature may make it less accessible to the average microbiome researcher. However, it is an important step in filling research gaps pertaining to underrepresented microbiomes in particular.

We thank the reviewer for their enthusiasm for our work.

R2.2) The paper is well-written, concise, and contains benchmarking steps that would be expected with this type of research. Benchmarking is done on a wide range of datasets, from simple mock communities, to non-human animal microbiomes and environmental samples. However, given the inherent challenges of testing metagenomic samples in the absence of a clear ground truth, the manuscript could benefit from further expanding the benchmarking in simulated metagenomes. For instance, the rationale behind the specific percentage of contaminant reads spiked in for Figure 2A is not clearly motivated, and it would strengthen the analysis to test multiple percentages. In addition, running multiple, independent simulation replicates for each data point and setting (in both Figures 2A and 2C) would improve the statistical robustness of the results and provide greater confidence in the reported performance.

Further benchmarks have been added to Figure 2 (now Figure 1, copied below), including the utilisation of multiple percentages of spiked-in reads from Eukaryotic sources, as well analysis of further CAMI 2 datasets. SPF performs as expected in these benchmarks, confirming our original conclusions about SPF's accuracy. However, we have not

included further benchmarking data for subpanel C, which assessed the accuracy of SPF in terms of increasing community novelty. Since this benchmark already includes 11 data points, we suggest that further replication would not substantively add to the benchmark's utility.

R2.3) The authors highlight instances where their tool is likely to give false estimates (e.g., microbiomes containing highly-abundant novel lineages or when genome size is vastly different than expected). In these cases, they have assured that the tool will warn the user of its potential pitfalls. The authors also made it clear that their tool will not necessarily provide better results than existing tools (e.g., STAT) for human fecal samples from well-studied populations. For this comparison, a short statement that the authors' results are not necessarily surprising is warranted since this approach is quite different from mapping to reference genomes. For future users, it would be beneficial for the authors to clearly state early on in the manuscript what the input formats should be for their tool. Does SMF only work in integration with SingleM? Highlighting how dependent this tool is on pre-existing tools from the authors would be beneficial so that users can understand how easily it can be integrated into their existing pipelines.

We thank the reviewer for this insightful suggestion. Indeed, SMF (now SPF) can be used in conjunction with other tools. We have added the below to the top of the methods section explaining this, copied below:

The SPF approach is presented here as a post-processing of SingleM-generated community profiles, but SPF can also be applied to community profiles generated by

other metagenome-based profiling tools, so long as they (1) estimate lineage-wise read coverage rather than lineage-wise read percentage, and (2) incorporate novel species into their profiles. The input to SPF is a "SingleM condensed" format file to specify the community profile (a simple sparse format readily generated from other community profile formats) and a count of total bases in the metagenome (alternately, if the metagenome itself is supplied, SPF will count the bases).

Minor comments:

R2.4) The authors can consider adding some numbers to support their statement in lines 318-319.

These lines refer to our analysis suggesting that human host DNA is present in publicly available metagenomes. We inspected a subset of these to confirm that the lower SPF values were not due to e.g. viral preparations or other library preparation methodologies which might skew the analysis, largely by interrogating whether the STAT analyses suggested substantial human DNA was present. However, we have not undertaken a comprehensive analysis because our analysis was not entirely automated. We are therefore not currently in a position to put hard numbers on the number of samples with human DNA in them without expanding the scope of the current manuscript. We also feel that highlighting specific instances of this may exacerbate the potential ethical implications of these observations, and so prefer not to report specific examples of metagenomes containing human DNA.

R2.5) The authors can consider merging the paragraph in lines 262-266 with that of lines 267-278.

These paragraphs, which both concerned analyses of marine systems, have been merged as suggested by the reviewer.

R2.6) Lines 292 and 293 both contain "see methods". Consider keeping only one such instance.

The second instance has been removed.

R2.7) Figure 1 is not the strongest figure and its message can easily be conveyed in the text. To remove or strengthen it is left to the authors.

This figure has been removed.

Re: mSystems01062-25R1 (Large-scale estimation of bacterial and archaeal DNA prevalence in metagenomes reveals biome-specific patterns)

Dear Dr. Ben Woodcroft:

Sorry for the delay in assessing your revision. Your manuscript has been accepted, and I am forwarding it to the ASM production staff for publication. Your paper will first be checked to make sure all elements meet the technical requirements. ASM staff will contact you if anything needs to be revised before copyediting and production can begin. Otherwise, you will be notified when your proofs are ready to be viewed.

Cover Image Submissions: If you would like to submit a potential Cover Image, please email a file and a short legend to mSystems@asmusa.org. Please note that we can only consider images that (i) the authors created or own and (ii) have not been previously published. By submitting, you agree that the image can be used under the same terms as the published article. Image File requirements: TIF/EPS, 7.5 inches wide by 8.25 inches tall (at least 2,250 pixels wide by 2,475 pixels tall), minimum 300 dpi resolution (600 dpi preferred), RGB, and no figure elements, e.g., arrows or panel labels. The legend should be a short description of the image, 1-2 sentences recommended. Please download and use this interactive template in Adobe to ensure that your proposed cover image meets our size requirements (<https://journals.asm.org/pb-assets/pdf-text-excel-files/ASM-Interactive-Sizing-Cover-Template-1715689791.pdf>).

Sincerely,
Sean Gibbons
Editor
mSystems

Reviewer #1 (Comments for the Author):

Thank you for the revisions! I have no further critiques to offer.

Reviewer #3 (Comments for the Author):

Thank you for the revisions. I appreciate the additional benchmarking and the clarified methodological description. The explanations provided for points where further changes were not made (e.g. human DNA quantification) are reasonable. I am satisfied that our concerns have been adequately addressed, and I have no further comments.